# Cell Membrane-Integrated Neuroligin-1 Regulates the Anti-Inflammatory Effects of CRC Cell-Derived Exosomes

**DOI:** 10.3390/ijms26020503

**Published:** 2025-01-09

**Authors:** Mohammad Mahmoudian, Francesco Trotta, Stefania Raimondo, Federico Bussolino, Marco Arese

**Affiliations:** 1Department of Oncology, University of Torino, 10060 Candiolo, Italy; mohammad.mahmoudian@unito.it; 2Candiolo Cancer Institute, FPO-IRCCS, 10060 Candiolo, Italy; 3Department of Chemistry, University of Torino, 10125 Torino, Italy; francesco.trotta@unito.it; 4Department of Clinical and Biological Sciences, University of Torino, Regione Gonzole 10, 10043 Orbassano, Italy; stefania.raimondo@unito.it; 5Neuroscience Institute Cavalieri Ottolenghi, University of Torino, Regione Gonzole 10, 10043 Orbassano, Italy

**Keywords:** Neuroligin-1, exosome, macrophages, colorectal cancer cells

## Abstract

Tumor-associated macrophages (TAMs) are one of the most abundant cell types in the colorectal cancer (CRC) tumor microenvironment (TME). CRC cell-derived exosomes support macrophage polarization toward an M2-like phenotype, which leads to tumor growth and metastasis. Neuroligin 1 (NLG1) is a transmembrane protein critical in synaptic function. We reported that NLG1 via an autocrine manner promotes CRC progression by modulating the APC/β-catenin pathway. This study aimed to answer whether NLG1 is involved in the exosome-mediated intercellular cross-talk between CRC and TAMs. Our results showed that exosomes of NLG1-expressing CRC cells induce M2-like (CD206^high^ CD80^low^) polarization in macrophages. On the other hand, we found that the exosomes of the NLG1 knocked-down CRC cells reinforce the expression of CD80 and pro-inflammatory genes, including IL8, IL1β, and TNFα, in the macrophages, indicating an M1-like phenotype polarization. In conclusion, NLG1, as a cell-membrane-integrated protein, could be a therapeutic target on the surface of the CRC cells for developing clinical treatments to inhibit exosome-induced anti-inflammatory immune responses in TME.

## 1. Introduction

Neuroligin-1 (NLG1) is a type 1 transmembrane protein consisting of a short cytoplasmic domain, a transmembrane part, and a large extracellular domain [1]. NLG1 is involved in forming excitatory synapses by binding to a presynaptic ligand, neurexin [2,3]. Connecting NLG1 with tumor progression and metastasis, our previously published study demonstrated that NLG1 is expressed in the clusters of aggressive migrating single tumor cells and vascular emboli of CRC and induces cell extravasation and organ metastatization [4]. Furthermore, we found that NLG1 promotes APC localization to the cell membrane and stimulates β-catenin translocation to the nucleus, upregulates mesenchymal markers and WNT target genes, and induces an epithelial–mesenchymal transition (EMT) in CRC cell lines [4]. These results indicated that NLG1, in an autocrine manner, promotes CRC development and metastasis. As the next step for unraveling the biological role of NLG1 in CRC development, we focused on the paracrine role of NLG1 via CRC-derived exosomes. Exosomes are critical mediators in intercellular communication in the tumor microenvironment (TME) [5]. These nanosized vesicles are released by most cells with a diameter of 30–150 nm and facilitate the transfer of diverse biomolecules such as proteins, lipids, amino acids, nucleic acids, and metabolites from cells of origin to recipient cells [6,7,8]. Macrophages are one of the most abundant immune cell types present in the TME, in which they are termed tumor-associated macrophages (TAMs). It is reported that exosome-mediated cross-talk between CRC cells with TAMs is pivotal in cancer cell invasion and migration [7,9]. TAMs can be categorized into two main subtypes; antitumor classically activated M1 phenotype and pro-tumor alternatively activated M2 phenotype [10,11]. Emerging evidences demonstrated that cancer cell-derived exosomes direct TAM polarization toward the M2-like phenotype [7,10,12,13]. M2 macrophages employ various anti-inflammatory routes to aid cancer cells throughout their metastasis, angiogenesis, and proliferation [14].

## 2. Results

### 2.1. NLG1 Expression in CRCs

Two categories can be distinguished between CRC cell lines based on the degree of NLG1 expression; one in which NLG1 was very low or absent named “NLG1 null” (such as HCT8, HT-29, and HCT116) and another constituted by the “NLG1 upregulated” (such as NCI-H716, MDST8, SNU-C2A, COLO320DM, HuTu 80, SNU-175, SNU-503) [4]. In this study, we selected HuTu 80 cells to investigate the regulatory role of NLG1 in the exosome-mediated cross-talk between TAMs and CRCs. In this study, NLG1 expression was downregulated using the NLG1-Human-pGFP-C-shLenti vector. The vector transduction was confirmed using fluorescent microscopy after drug selection (Puromycin) of the transduced cells (Figure 1a). Then, the mRNA expression level of NLG1 was analyzed using qRT-PCR (Figure 1b), demonstrating an 8-time downregulation after viral transduction. In addition, the intensity of the immunoblot band of NLG1 was decreased after viral transduction. An immunoprecipitation assay was performed using an anti-NLG1 (N97A/31) antibody to recognize the location of NLG1 in WB papers (Figure 1c).

### 2.2. Characteristics of the Exosomes

Exosomes were isolated using the ultracentrifuge method (Figure 2a) and characterized for morphology, size, surface charge, and exosomal markers using Transmission Electron Microscopy (TEM), Nano Tracking Analysis (NTA), Malvern Zetasizer, and WB assay, respectively. The TEM images demonstrated a spherical morphology for the exosomes (Figure 2b) and NTA data revealed a hydrodynamic diameter mode of 124 nm (SD = 11, PDI = 0.008) (Figure 2c). A negative zeta potential value of −13 mV (SD = 1, PDI = 0.006) was obtained for the exosome (Figure 2d). The presence of CD 63 as an exosomal protein maker was confirmed by the WB assay, while calnexin (exosome absence marker) was not detected in the exosomes (Figure 2e).

### 2.3. The Cell Adhesion and Morphology of the THP-1 Macrophages Depend on the PMA Concentration in a Dose-Dependent Manner

Inflammatory (M1) and anti-inflammatory (M2) macrophages are two main types of TAMs. TAMs can be obtained in vitro firstly by the polarization of the human monocytic leukemia cell line (THP-1) into undifferentiated macrophages (M0) using phorbol 12-myristate 13-acetate (PMA) as a primary stimulus [15]. Then, M0 macrophages were treated with secondary stimuli; IFN-γ/LPS or IL-4/IL-13 to induce M1 or M2 phenotypes, respectively. In this study, the optimized macrophage polarization condition to have M0 macrophages was considered as: 1 × 10^6^ THP-1 cells/well of 6-well culture plate, 24 h treating time with 2 mL/well of PMA (100 ng/mL in RPMI culture media), followed by 24 h resting time. M0 macrophages were treated with exosomes or secondary stimuli of M1/M2 macrophages for 48 h. The total period of experiments should not exceed more than three days after removing PMA from culture media. Our investigation revealed that three days after removing PMA from the culture media, the vast majority of adherent M0 macrophages detached from the plate; this took place more significantly for the THP-1 cells that were treated with lower concentrations of PMA (5 and 25 ng/mL) (Figure 3b). Therefore, we considered one day for M0 macrophages resting after removing PMA and two days for the treatment period of the rested-M0 macrophages with different types of secondary stimuli. We also checked the density of THP-1 cells (250,000, 500,000, and 1,000,000 cells/well), the volume of the PMA-treating solution (1 mL vs. 2 mL), and PMA concentration (5, 25, and 100 ng/mL) on the adhesion density of M0 macrophages. Cell adhesion and spreading are hallmarks of monocyte polarization into M0 macrophages. We found that, when the THP-1 cells were treated with a 1 mL/well of PMA, the cell adhesion density was dramatically less than the treating volume of the 2 mL/well (Appendix A). The optimum density of THP-1 cells for macrophage polarization was obtained at 1,000,000 cells/well (Appendix A).

Different morphology of M0 macrophages are depicted in Figure 3a, including rounded appearance, flattened appearance, ”Fried egg” morphology, and ”Enlongated spindle-like” morphology [16]. Most importantly, our results indicated that the concentration of PMA is very important in the cell attachment duration and the morphology of M0 macrophages. We observed that after three days of removing PMA, the majority of M0 macrophages detached from the culture plates when we used lower concentrations of PMA (5 and 25 ng/mL) for THP-1 polarization (Figure 3b). Additionally, we noticed that, at a lower treatment concentration of PMA (5 ng/mL), the macrophage morphology was a mix of rounded appearance, flattened appearance, and ”Fried egg” appearance, while at the higher concentration (100 ng/mL), the ”Fried egg” and ”Enlongated spindle-like” morphologies were the predominant ones. When PMA was removed from the culture media, the morphology of the M0 macrophage started to turn into a rounded appearance and then lost contact with the culture plate.

### 2.4. High PMA Concentration Augments the Expression of the Pro-Inflammatory Chemokine/Cytokine Genes in THP-1 Macrophages

In the next step for optimizing the macrophage polarization condition, we investigate the impact of the PMA concentration and macrophage resting on the PMA-induced gene expression in the M0 macrophages. We selected 11 genes including M1-related genes (CD80, CD86, IL8, IL6, IL1β, TNFα, CCL2, and CCL5) and M2-related genes (CD206, CD200R, and CCL22) and quantified by qRT-PCR. These genes were selected according to the recently published literature [11,15,17,18,19,20,21]. We polarized THP-1 cells (1,000,000 cells/well in a 6-well culture plate) by treating them with 2 mL of PMA (25 and 100 ng/mL). As shown in Figure 4, CCL5 and IL1β genes were highly expressed in the M0 macrophages. Macrophage resting after polarization and, most importantly, using a lower concentration of PMA, decreases the pro-inflammatory cytokine transcriptional expression. The mRNA gene expression of CCL5, IL1β, TNFα, and IL8, were downregulated 9 times, 127 times, 7 times, and 8 times, respectively, by decreasing the PMA concentration from 100 ng/mL to 25 ng/mL. Intriguingly, the expression level of the CD206 gene, M2 marker, increases by 2.5- and 13-time follow-up resting cells after PMA stimulation and decreasing PMA concentration from 100 to 25 ng/mL, respectively.

### 2.5. Exosomes of NLG1-ExpressingCRC Cells Induce M2-like Macrophage Polarization

The crosstalk between tumor cells and TAMs actively participates in tumor progression and metastasis. Exosomes are considered important mediators of this intercellular communication [6]. As an example, miR-151-3p derived from gastric cancer exosomes induces M2-phenotype polarization of macrophages and promotes tumor growth [22]. Macrophage polarization can be assessed by analyzing the cell marker expression level or cytokine/chemokine secretion pattern. The impact of the exosomes of NLG1-expressing CRC cells on the marker expression of the M0 macrophages was assessed by analyzing the mRNA expression level of CD206 and CD80 genes as markers for M2 and M1 phenotypes, respectively. M0 macrophages treated with IL13+1L4 (M2-stimuli) and LPS+INF-γ (M1-stimuli) were considered controls for M2- and M1-like phenotypes, respectively. Both the exosomes and IL13+1L4 stimuli upregulated the expression of the CD206 gene and suppressed the expression of the CD80 gene (Figure 5a). The genes that were chosen to examine the effect of exosomes on the expression of pro-inflammatory cytokines/chemokines were IL8, IL6, TNFα, and IL1β. As shown in Figure 5b, the mRNA expression level of the pro-inflammatory cytokines/chemokines was downregulated in M0 macrophages exposed to the exosomes and IL13+1L4 stimuli. Our results demonstrated that the exosomes of NLG1-expressing CRC cells direct macrophage polarization toward an M2-like phenotype.

### 2.6. Exosomes of NLG1 Knock-Down CRC Cells Augment Macrophage Polarization Toward M1-like Phenotype

M0 macrophages were treated with the exosomes of the NLG1-expressing/knock-downed CRC cells. Intriguingly, we found that knock-downing NLG1 protein in the CRC cells suppressed the anti-inflammatory effect of the exosomes on the macrophages. Exosomes of the NLG1 knocked down CRC cells remarkably increased the expression level of the CD80 gene and reduced the expression level of the CD206 thus supporting the M1-like phenotype of the macrophages (Figure 6a). Meanwhile, we found that the exosomes of the NLG1 knocked-down CRC cells augment the expression of pro-inflammatory genes including IL8, IL1β, and TNFα (Figure 6b).

## 3. Discussion

THP-1 monocytes have been used as an in vitro macrophage model. Of the various agents capable of inducing the differentiation of THP-1 monocytes to macrophages, PMA has been predominantly used at 5–100 ng/mL concentrations with or without resting. Simple checks of cell adhesion and morphology, as well as an analysis of surface marker expression, have been used to confirm the differentiation process. In this study, we applied different parameters including the PMA concentration, the density of THP-1 monocyte cells, and macrophage resting to optimize THP-1 differentiation into M0 macrophages. We discovered that the length of time macrophages remain attached to the culture plate surface is highly influenced by the PMA concentration. Macrophages that are heavily attached are produced by higher PMA concentrations. When PMA was removed from the culture media, the morphology of the macrophages started to change from flattened/elongated spindle-like morphology into monocyte round shape, and the cells lost their ability to adhere. However, the transcriptomic analysis revealed that the gene expression of the M0 macrophages, especially pro-inflammatory genes, is dramatically influenced at high PMA induction concentrations. In particular, CCL5 and IL-1b were highly activated genes at a high concentration of PMA (100 ng/mL vs. 25 ng/mL), which can mask the effect of the secondary stimuli on the expression level of the respected genes in the M0 macrophages. Aligned with our results, Park, E. K. et al. [15] reported that the PMA increases cytokine expression dose-dependently. Furthermore, we found that macrophages resting followed PMA induction and, more importantly, when PMA concentrations were decreased, there was an increase in the expression of CD206, a phenotypic marker for M2-like TAMs. Consistent with our results, Baxter, E. W. et al. [17] reported that M0 macrophage resting after PMA treatment allows for the transcription of M2 marker genes. In this study, we examined the expression level of several pro/anti-inflammatory genes for two main purposes: optimizing the macrophage polarization state was the initial step, followed by choosing the right genes for the characterization of the polarized M1/M2 macrophages. For gene selection, we took into account the criteria that pro-inflammatory genes should have high expression in polarized M1 macrophages as opposed to M0/M2 phenotypes. In contrast, the anti-inflammatory genes should have high expression levels in polarized M2 macrophages compared with M0/M1 phenotypes.

Following the polarization process described above, it was discovered that, aside from the genes shown in Figure 4, the CD163, TGFβ, IL10, and IL12b genes could not be regarded as distinctive genes for M1/M2 phenotypes according to our gene selection criteria. Compared to M1 macrophages, M2/M0 macrophages had higher levels of IL12 gene expression (an M1 marker), whereas M1/M0 macrophages demonstrated higher gene expression levels of CD163, TGFβ, and IL10 (an M2 marker) compared with M2 macrophages (data not reported here).

Functional and phenotypic plasticity is a typical feature of TAMs. Depending on signaling stimulations in the TME, the phenotype of TAMs will be adopted into a pro-inflammatory (M1) to anti-inflammatory (M2) state [23,24]. It is reported that CRC cell-derived exosomes are important stimuli in TME, which induce M2 polarization of macrophages resulting in tumor progression and metastasis [7,25,26]. Underlying the intracellular mediators that regulate the exosomal packaging of the cytoplasmic biomolecules are still poorly elucidated. Future progress in this field may open new windows to develop clinical targets to modulate the exosome-mediated paracrine activity of the cancer cells in reshaping TME. For example, Chairoungdua, A., and et al. [27] reported that CD89 and CD9 are tetraspanin membrane proteins that can suppress tumor metastasis. They explained that CD82 and CD9 expression induces β-catenin export via exosomes, which is blocked by a sphingomyelinase inhibitor (GW4869).

Our recently published paper showed that the neuronal protein NLG 1 promotes CRC progression by modulating the APC/β-catenin pathway [4]. This study aimed to answer whether NLG1 can be a cell membrane-integrated protein that controls the exosomal packaging of cytoplasmic biomolecules in CRC cells. For this aim, we first assessed the effect of exosomes of NLG1-expressing CRC cells on the M0 macrophage polarization by analyzing the mRNA expression level of macrophage markers (CD206, CD80) and pro-inflammatory cytokines/chemokines including IL8, IL6, IL1β, and TNFα. We found that the exosomes of NLG1-expressing CRC cells remarkably suppress the expression of the pro-inflammatory cytokines/chemokines, which was more significant for IL1β and TNFα genes. IL-1β and TNF-α are functional markers of M1 macrophages. M1 macrophages are characterized by their ability to produce high levels of pro-inflammatory cytokines, including IL-1β and TNF-α, which play critical roles in the inflammatory response and are representative of an activated pro-inflammatory state [28]. In the meantime, the exosomes phenotypically direct macrophages toward CD206^high^ CD80^low^. CD80 is considered one of the key markers of M1 macrophages, with the power to discriminate between M1 and M2 polarization states [19]. In contrast, M2 macrophages typically express CD206, also known as mannose receptor C type 1 (MRC1). CD206 is one of the most commonly used markers for identifying M2 macrophages that play a critical role in anti-inflammatory responses [29]. In the next step, NLG1 was silenced to find out whether this protein regulates the anti-inflammatory effects of the exosomes or not. We found that the exosomes of NLG1-silenced CRC cells direct M0 macrophages toward CD206^low^ CD80^high^ phenotype. Meanwhile, the expression level of IL8, IL1β, and TNFα genes dramatically increased in the exosome-treated macrophages, indicating an M1-like phenotype polarization. While our findings provide compelling evidence of macrophage polarization influenced by exosomes from CRC cells, as demonstrated by gene expression changes and surface marker profiles, the lack of functional validation (e.g., phagocytic ability or cytokine secretion analysis) is a limitation of our study. Future work will focus on integrating these functional assays to comprehensively characterize the phenotypic and functional states of polarized macrophages. In conclusion, NLG1 as a cell-membrane integrated protein could be a therapeutic target on the surface of the CRC cells to inhibit the anti-inflammatory responses of the macrophages initiated by the exosomes of the CRC cells. To elaborate on the regulatory role of the NLG1 in the cytoplasmic biomolecules loading inside the exosomes, our future study will concentrate on complementary investigations such as proteomic and high-throughput transcriptome analysis of the exosomes.

The observed influence of NLG1 on macrophage polarization raises intriguing questions about the specific mechanisms underlying exosome packaging and transcellular signaling. NLG1 may regulate the loading of specific molecular cargo into exosomes, such as miRNAs or protein factors, which in turn drive M2- or M1-like polarization. For example, NLG1 might facilitate the inclusion of anti-inflammatory miRNAs or immunosuppressive proteins in exosomes from NLG1-expressing CRC cells, while its knockdown may alter this cargo toward a pro-inflammatory profile. Furthermore, it is plausible that NLG1-mediated exosome signaling establishes a feedback loop within the tumor microenvironment (TME), influencing not only macrophage behavior but also other immune and non-immune cells. Such a feedback regulation mechanism could amplify the immunomodulatory effects of NLG1 in the TME. Future studies should aim to delineate the specific cargo and signaling pathways regulated by NLG1 in exosomes to unravel these complex interactions.

## 4. Methods and Materials

### 4.1. Cell Lines

All CRC cell lines used in this paper are part of the Candiolo Institute cell bank. HT-29 (ATCC), 293 T cells, THP-1 (ATCC), and HuTu 80 (ATCC) cells were grown in DMEM-High glucose, DMEM-High glucose, RMPI 1640, and MEM culture media, respectively All culture Media were provided from Euroclone S.p.A. (Pero, MI, Italy), and supplemented with 10% FBS (heat-inactivated, Euroclone S.p.A., Pero, MI, Italy), 1% antibiotic (penicillin-streptomycin, Euroclone S.p.A., Pero, MI, Italy), and 1% glutamine (Euroclone S.p.A., Pero, MI, Italy). All cell lines were regularly tested for mycoplasma with Venor^®^ GM Kit (Minerva Biolabs, Berlin, Germany).

### 4.2. Transient Transfection of 293T Cells for Lentiviral Vectors Production

The 293T cells were seeded in a 15 cm dish. The 80% confluent cells were sub-cultured into sex culture dishes (15 cm) for 24 h before transfection in 20 mL final volume cell culture media. Then, the culture media was replaced with complete fresh media (DMEM, 10% heat-inactivated FBS, 1% glutamine, 1% antibiotic) 2 h before transfection. The plasmid DNA mixture was prepared for one dish by adding: enveloping plasmid (VSV-G, 9 µg), packaging plasmid (pMDLg/pRRE, 12.5 µg), REV plasmid (6.25 µg), and transfer vector plasmid (NLGN1-Human-pGFP-C-shLenti, 32 µg). The plasmid solution was made up to a final volume of 1125 µL with 0.1× TE/dH_2_O (2:1); for preparing 0.1× TE buffer: 10 mM Tris (pH 8.0) and 1 mM EDTA (pH 8.0) was diluted at a ratio of 1:10 with sterilized dH_2_O. Finally, 125 µL (250 µL) of 2.5 M CaCl_2_ was added into the plasmid mixture and incubated at room temperature for 5 min. The precipitate was formed by dropwise addition of 1250 µL of HBS solution (2×, 281 mM NaCl, 100 mM HEPES, 1.5 mM Na_2_HPO_4_, pH 7.12) to the equal volume of DNA-TE-CaCl_2_ mixture, while vertexing at full speed. The final solution was added to the 293T cell culture media immediately. A very small granular precipitate of the CaPi-precipitated plasmid DNA was observed above the cell monolayer using microscopy. The transfected cells were incubated for 16 h at 37 °C and then the media was replaced with 16 mL fresh media. After 24 h incubation at 37 °C, the culture media was collected and filtered (0.2 µm). For virus collection, the filtered media was centrifuged at 19,500 rpm for 130 min at 18 °C. The virus pellet was dispersed in sterile PBS, quantified with a P24 test, and stored in a freezer at −80 °C.

### 4.3. Cell Transduction with sh-RNA-NLG1 Lentivirus

NLG1 silenced HuTu80 cell populations were generated by infection with lentiviral vectors targeting NLG1 from Origene: cat # TL3111634B (shNLGN1b) (pGFP-C-shLenti shRNA-29mer expression vector-ORIGENE, Rockville, MD, USA). HuTu80 cells were seeded in the 6-well plate (200,000 cells/well). The next afternoon, the culture media was replaced with 1 mL transduction solution per each well; 990 µL (cell culture media, supplemented with 10% FBS (heat-inactivated), 1% glutamine, 1% antibiotic) + 1 µL concentrated virus solution (concentration of virus was 60 ng) + 1 µL polybrene (1000×; 8 mg/mL, Lot. 210929L01, VectorBuilder, Chicago, IL, USA). The next day morning, the transduction media was replaced with 2 mL of fresh complete media. When the cell confluence of the transduced cells reached 80%, the cell was treated with puromycin (2 mg/mL, 1 µL in 1 mL culture media) for 48 h for drug selection. The efficiency of NLG1 expression downregulation was checked by qRT-PCR analysis and Western blot assay.

### 4.4. Exosome Isolation and Characterization

Exosomes were isolated from cell culture media using the ultracentrifuge method. In brief, the cells were cultured in a T75 cm^2^ cell culture flask. When the cell confluence reached 80%, the cell culture media was replaced with complete media (2% exosome-free FBS, 1% antibiotic, and 1% glutamine), and the cells were incubated for 48 h at 37 °C. Then, the culture media were collected and centrifuged at 4 °C and 300× *g* for 10 min, 2000× *g* for 20 min, and 20,000× *g* for 10 min. Finally, the culture media were collected and ultracentrifuged at 4 °C and 110,000× *g* for 70 min (Optima™ L.100 XP Ultracentrifuge, BECKMAN COULTER, Brea, CA, USA). In the end, the exosomes pellet was dispersed in sterile PBS and sorted in a −80 °C freezer for the next experiments. Exosomal protein content was quantified with a BCA Protein Assay Reagent Kit (Pierce Chemical Co., Dallas, TX, USA). The exosomes were characterized for morphology, size, surface charge, and exosomal markers using Transmission Electron Microscopy (TEM; JEM-1010, JEOL, Tokyo, Japan), Nano Tracking Analysis (NTA; NS300, NANOSIGHT, Malvern, UK), Zetasizer (Malvern Instruments, Malvern, UK), and Western blot (WB), respectively. Exosome samples (with protein content of 280 μg/mL) were diluted 1:20 with PBS and vortexed for 2 min before TEM and NTA. The exosome solution was not diluted for the size and zeta potential analysis.

For TEM imaging, 10 µL of the diluted exosome solution was placed on the shiny side of the grid (Pioloform^®^ coated grids, Mesh cu, S134, Agar Scientific, Rotherham, UK) and dried at room temperature. For washing, water droplets were placed on the parafilm sheet, and then the grid surface contact on the top of the water droplets (4 times). The residual water was removed by a paper filter. Then, the shiny side of the grid was contacted with a drop of uranyl acetate solution (4%, prepared in water), which was placed on the parafilm sheet, for a few seconds (e.g., 30 s). The washing step was repeated 4 times and the residual water was removed by a paper filter. The grids were dried at room temperature overnight in a dark place and then observed under the electron microscope at 80 kV.

### 4.5. Macrophage Polarization

The human monocytic leukemia cell line, THP-1 cell line, was polarized into macrophage-like state (M0) by treating THP-1 monocytes for 24 h with 100 ng/mL of phorbol 12-myristate 13-acetate (PMA; P1585, Sigma, St. Louis, MO, USA) in 6-well cell culture plates with working solution volumes of 2 mL and a primary cell density of 1 × 10^6^ cells/well. Differentiated, adherent cells were washed twice with culture medium (RPMI 1640 medium) and rested for another 24 h in the completed culture medium to obtain the resting state of macrophages (M0). Then, the M0 macrophages were treated with fresh medium supplemented with 20 ng/mL IFN-γ (ab259377, Abcam, Waltham, MA, USA) + 100 ng/mL LPS (L2630, Sigma) to differentiate into the M1 phenotype and with 20 ng/mL IL-4 (P3682, Abnova, Littleton, CO, USA) + 20 ng/mL IL-13 (ab270079, abcam) to the M2 phenotype, and 50 μg/mL of exosomes for exosome-treated M0 macrophages. The incubation time was 48 h in all stimulating conditions.

### 4.6. Immunoprecipitation, Immunoblotting Analysis

The cells were cultured in a 6-well cell culture plate (250,000 cells/well). Then, subconfluent cells were homogenized in 300 μL/well cold lysis buffer (10 mM Tris-HCl, pH 7.5, 150 mM NaCl, 5 mM EDTA, pH 8; 10% glycerol, 1% Triton X-100) supplemented with protease inhibitor 1000×, phosphatase inhibitor cocktail II (100×, MCE), 1 mM PMSF, and 20 mM sodium pyrophosphate. The lysates were collected into a 1.5 mL tube and centrifuged at 10,000× *g* for 20 min (4 °C). Afterward, the supernatants were collected and quantified with a BCA Protein Assay Reagent Kit (Pierce Chemical Co., Dallas, TX, USA). Then, 1 mg of total lysate was incubated overnight at 4 °C, with gentile agitation, with primary mouse monoclonal anti-Neuroligin 1 antibody (N97A/31, NeuroMab, Davis, CA, USA) used at a final concentration of 2.5 μg/1 mg of protein. To recover the immune complexes, 100 μL of protein A-Sepharose beads (Protein A Sepharose™ CL-4B, Cytiva, Marlborough, MA, USA) prepared in lysis buffer was added into each lysate and incubated at 4 °C for 90 min with gentle shaking. The beads were pelleted and washed 3 times with lysis buffer at 1000× *g* for 5 min. After the fourth wash, the supernatant was aspirated and the beads were resuspended in 50 μL of LDS sample buffer (1×, NuPAGE™, Invitrogen, Waltham, MA, USA) and boiled at 95 °C for 8 min. At the end, the sample buffer was centrifuged at 10,000× *g* for 1 min at 4 °C and the protein solution was analyzed by SDS-PAGE electrophoresis gel, using nitrocellulose (NC) membrane. A protein content of 10 μg in a final volume of 38 μL was loaded into each well. For Western blotting analysis, NC membranes were immunodecorated with primary antibodies; anti-Calnexin (AF18, sc-23954), anti-CD63 (MX-49.129.5, sc-5275), and anti-Neuroligin 1 (N97A/31, NeuroMab, Davis, CA, USA). Secondary antibodies were Multi-rAb HRP-Goat Antibody (Proteintech^®^, Rosemont, IL, USA), which were detected with ECL reagent (Amersham/GE Healthcare, Amersham, UK).

### 4.7. RNA Extraction, Retrotranscription, and qRT-PCR

According to the manufacturer’s instructions, the total RNA was extracted using Maxwell^®^ RSC miRNA Tissue Kit (AS1460, Promega, Madison, WI, USA). RNA purity was checked using the NanoPhotometer1 spectrophotometer (IMPLEN, München, Germany), and RNA integrity was assessed using the RNA 6000 Nano Kit COD.5067-1511 of the BioAnalyzer 100 system (Agilent Technologies, Santa Clara, CA, USA). First-strand cDNA was generated from 1 μg of total RNA using the iScriptTM Reverse Transcription supermix for RT-qPCR Kit (Bio-rad, Hercules, CA, USA). The expression of human NLG1 (Hs00208784_m1), CD86 (Hs01567026_m1), CD80 (Hs00175478_m1), MRC1 (Hs00267207_m1), CD200R1 (Hs00990600_g1), TNFα (Hs00174128_m1), IL1β (Hs00174097_m1), IL8 (Hs00174103_m1), IL6 (Hs00174131_m1), CCL22 (Hs01574247_m1), CCL2 (Hs00234140_m1), and CCL5 (Hs00982282_m1), was analyzed by quantitative real-time reverse transcription-PCR (qRT-PCR) using a TaqMan Gene Expression Assay (Applied Biosystem, Waltham, MA, USA). The mRNA levels were analyzed in triplicate and were normalized against human TATA-binding box protein (TBP) (Hs00427620_m1).

## 5. Conclusions

Exosomes derived from CRC cells are significant stimuli, eliciting anti-inflammatory responses in TME and contributing to tumor development and metastasis. Underlying the cellular mediators that regulate the exosomal packaging of the cytoplasmic biomolecules may open new windows to develop clinical targets to modulate the exosome-mediated TME reshaping in cancer. Tumoral NLG1 is a cell membrane-integrated protein that promotes CRC progression in an autocrine manner. This study demonstrated that the exosomes of NLG1-expressing CRC cells reinforce anti-inflammatory responses and M2-like polarization in the macrophages. Meanwhile, NLG1-silencing diminished the anti-inflammatory effect of the exosomes by reinforcing the M1-like polarization in the macrophages. In summary, our study suggests that NLG1 as a cell-membrane-integrated protein could be a therapeutic target in CRC to suppress anti-inflammatory immune responses in TME initiated by exosomes of the CRC cells.

## Figures and Tables

**Figure 1 ijms-26-00503-f001:**
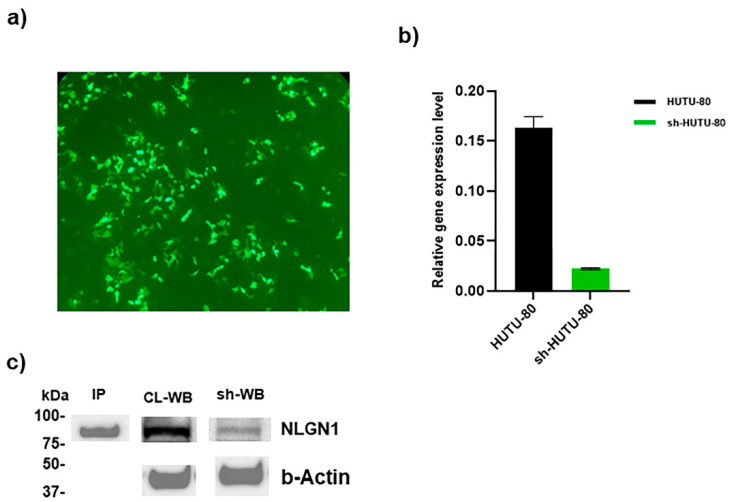
Schematic of the NLG1 downregulation using NLG1-Human-pGFP-C-shLenti vector. The viral vector transduction was monitored using fluorescent microscopy with a magnification scale of 10× (**a**) after drug selection of the transduced cells. The mRNA expression level of NLG1 was analyzed using qRT-PCR (**b**). Relative gene expression (2^−∆Ct^) was calculated relative to TBP expression. Data shown for qRT-PCR are the means ± standard deviation (SD). An immunoprecipitation assay was performed using an antiNLG1 (N97A/31) antibody to recognize the location of NLG1 in WB papers. A WB assay on the nitrocellulose papers was performed to evaluate NLG1 downregulation at the protein level (**c**).

**Figure 2 ijms-26-00503-f002:**
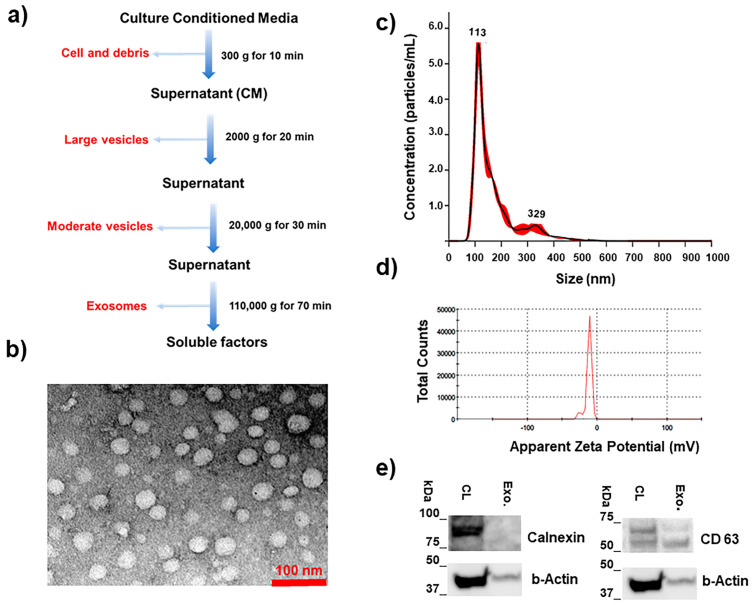
Exosome isolation using the ultracentrifuge method (**a**), and characterized for morphology, size, surface charge, and exosomal markers using: Transmission Electron Microscopy (TEM) (**b**), Nano Tracking Analysis (NTA) (**c**), Malvern Zetasizer (**d**), and WB assay (**e**), respectively.

**Figure 3 ijms-26-00503-f003:**
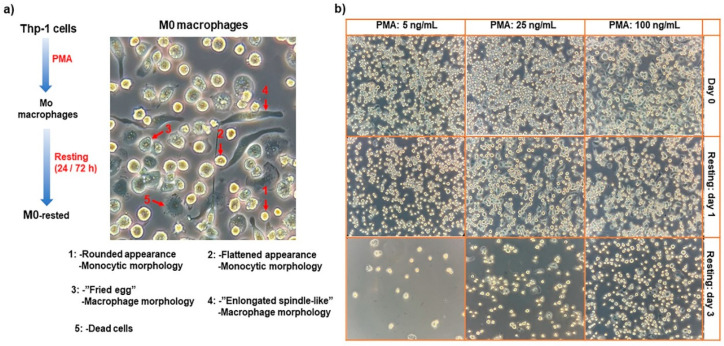
Visualized morphology of the differentiated human monocytes leukemia cell line, THP-1 cell line, into a macrophage-like state (M0), with a magnification scale of 20× (**a**). Effect of PMA concentration and resting time on the adhesion and spreading of the M0 macrophages, with a magnification scale of 10× (**b**).

**Figure 4 ijms-26-00503-f004:**
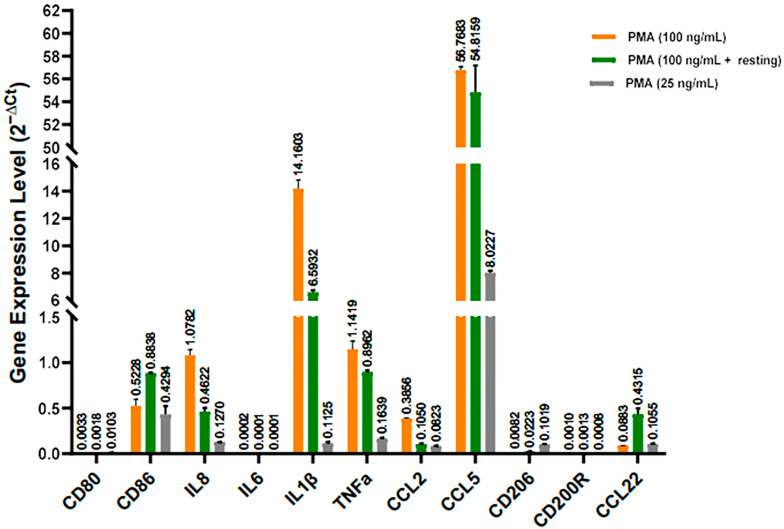
Effect of PMA concentration and resting time on the inductions of some cell markers, cytokines, and chemokines during THP-1 differentiation. Relative gene expression (2^−∆Ct^) was calculated relative to TBP expression. Data shown for qRT-PCR are the means ± standard deviation (SD).

**Figure 5 ijms-26-00503-f005:**
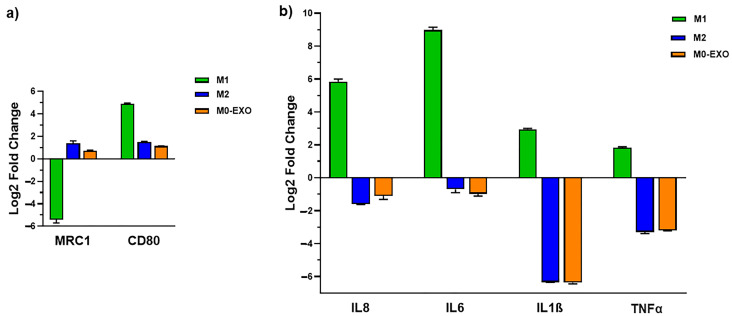
M0 macrophages were treated with exosomes of NLG1-expressing CRC cells (HuTu80 cells), IL13+1L4 (M2-stimuli), LPS+INF-γ (M1-stimuli) for 48 h. The expression levels of CD206, and CD80 as makers for the macrophage phenotype (**a**), IL8, IL6, TNFa, and IL1β genes as makers for the macrophage functionality (**b**), were determined using qRT-PCR. Gene expression level (Log2 FC) was calculated relative to untreated M0 macrophages. Data shown are the means ± standard deviation (SD).

**Figure 6 ijms-26-00503-f006:**
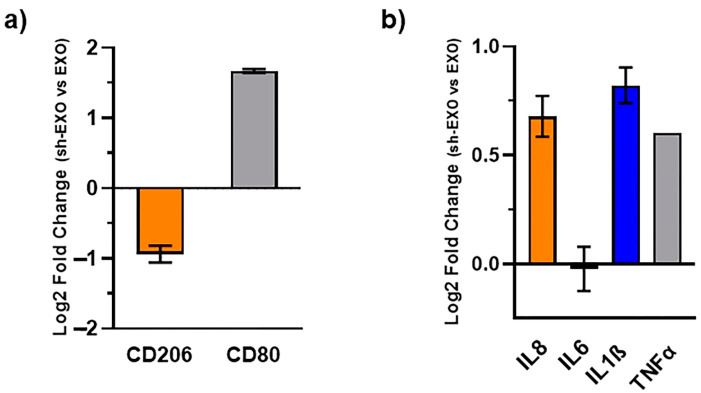
M0 macrophages were treated with the exosomes of the NLG1 expressing (EXO) and NLG1 knock-downed (sh-EXO) CRC cells and the mRNA expression level (Log2 FC) of the CD206 and CD80 genes as phenotypical markers (**a**) and pro-inflammatory genes including IL8, IL6, IL1β, and TNFa as makers for the macrophage functionality (**b**), were analyzed using qRT-PCR. Data shown for qRT-PCR are the means ± standard deviation (SD).

## Data Availability

Data is contained within the article and Appendix A.

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
