# Peer review of "Cell Membrane-Integrated Neuroligin-1 Regulates the Anti-Inflammatory Effects of CRC Cell-Derived Exosomes"

_ijms, 2025, doi:10.3390/ijms26020503_

Round 1
Reviewer 1 Report
Comments and Suggestions for Authors
This study comprehensively demonstrates that the exosomes derived from NLG1 ex-pressing-CRC cells are able to phenotypically and functionally direct macrophages toward an M2-like anti-inflammatory and protumorigenic phenotype, thus suggesting that NLG1 as a cell-membrane integrated protein could be a therapeutic target in CRC to suppress anti-inflammatory immune responses in TME initiated by exosomes of the CRC cells.
Results and the topic are very interesting. The manuscript is clearly written, methods are rigorous, conclusions are supported by results. Introduction and Discussion are well organized. The manuscript is also well referenced: I just suggest this Ref. (PMID: 36009243) if authors consider it interesting to add at row 46-47 of the Introduction. English is fine.
I have only a consideration to do, which is the follow. Typically, inflammation is considered a protumorigenic factor, while here, according to what you demonstrated, macrophages (THP-1), directed by NLG1 ex-pressing-CRC cells, differentiate towards an anti-inflammatory phenotype and this is associated with cancer progression, something that is apparently in contrast with the generally accepted role of inflammation in carcinogenesis and cancer progression. Aware of the fact that “each cancer tells its own story”, could you please make a comment on this? Thank you
Author Response
Comment 1: I just suggest this Ref. (PMID: 36009243) if authors consider it interesting to add row 46-47 of the Introduction.
Response: The reference is included in the introduction. Thank you for this suggestion.
Comment 2: I have only a consideration to do, which is the follow. Typically, inflammation is considered a protumorigenic factor, while here, according to what you demonstrated, macrophages (THP-1), directed by NLG1 ex-pressing-CRC cells, differentiate towards an anti-inflammatory phenotype and this is associated with cancer progression, something that is apparently in contrast with the generally accepted role of inflammation in carcinogenesis and cancer progression. Aware of the fact that “each cancer tells its own story”, could you please make a comment on this? Thank you
Response: Thank you for your insightful comment. You are absolutely correct that inflammation is typically considered a protumorigenic factor in cancer progression. However, as you noted, each cancer has its unique microenvironment, and the role of inflammation can vary depending on the specific context. In our study, we observed that the exosomes from neuroligin-1 (NLG1)-expressing colorectal cancer (CRC) cells directed the polarization of THP-1 macrophages toward the M2 phenotype. This observation aligns with the well-established concept that tumor-associated macrophages (TAMs) often adopt an M2-like phenotype in many cancers, including CRC. These M2 macrophages contribute to cancer progression by promoting extracellular matrix remodeling, angiogenesis, and immune suppression, which facilitate tumor growth and metastasis. While inflammation is indeed protumorigenic, it is important to differentiate between acute inflammation and the immunosuppressive environment created by chronic inflammation and M2 macrophage activity in the TME. Chronic inflammation can transition into an immunosuppressive state, where M2 macrophages suppress adaptive immune responses, thus aiding in tumor immune evasion and progression. This paradoxical role of inflammation, where the acute phase might initiate a protumorigenic environment but eventually shifts toward a chronic, anti-inflammatory state conducive to tumor progression, reflects the duality of the immune system's role in cancer.
Reviewer 2 Report
Comments and Suggestions for Authors
The manuscript reveals that exosomes of NLG1-expressing CRC cells reinforce anti-inflammatory responses and M2-like polarization in the macrophages, and NLG1-silencing diminished the antiinflammatory effect of the exosomes by reinforcing the M1-like polarization in the macrophages. This suggests that NLG1 as a cell-membrane integrated protein could be a therapeutic target in CRC to suppress anti-inflammatory immune responses in TME initiated by exosomes of the CRC cells. The work is novel and innovative, and it addresses the knowledge gap on role of antiinflammatory role of exosomes. References are adequate and up to date. I only have some minor comments to be addressed before the manuscript can be accepted for publication:
1- The introduction section, the conclusion of the paper needs to be removed from the end of the introduction section.
2- Section 2.1., the antibiotic needs to be specified, as well as its source and the source of glutamine.
3- Section 3.2, specify whether the exosomal samples were diluted before measuring the properties using zetasizer device.
4- Report the particle size value and zeta potential of exosomes as mean and S.D. Also, the PDI should be specified as well, as it's very important to indicate the homogeneity of the sample.
Author Response
COMMENT 1: The introduction section, the conclusion of the paper needs to be removed from the end of the introduction section.
Response: The conclusion paragraph has been removed from the end of the introduction section. Thanks a lot for this comment.
COMMENT 2: the antibiotic needs to be specified, as well as its source and the source of glutamine
Response: The source of the reagents was Gibco-Fisher Scientific and the type of antibiotics was penicillin-streptomycin. This information was included in the methodology section.
COMMENT 3: specify whether the exosomal samples were diluted before measuring the properties using zetasizer device.
Response: Thank you for your insightful comment. The exosome samples were just diluted for TEM and NTA analysis as we needed more diluted samples (the dilution ratio (1:20) was explained in the methodology section). In the methodology section, we added the point that the original exosome solution (280ug/ml) was not diluted for size/surface charge analysis using a zetasizer device.
COMMENT 4: Report the particle size value and zeta potential of exosomes as mean and S.D. Also, the PDI should be specified as well, as it's very important to indicate the homogeneity of the sample.
Response: Thank you for your insightful comment. SD and PDI values were added to the mean particle size and zeta potential values of exosome characterization. Mean particle size: 124 nm (SD= 11, PDI= 0.008), Zeta potential: -13 mV (SD=1, PDI= 0.006)
Reviewer 3 Report
Comments and Suggestions for Authors
The manuscript deeply explores the mechanism of action of NLG1 between colorectal cancer (CRC) cell-derived exosomes and tumor-associated macrophages (TAMs) polarization, and proposes the possibility of NLG1 as a potential therapeutic target to regulate immune response in the tumor microenvironment (TME). The manuscript systematically studies for the first time the mechanism of action of NLG1 in regulating CRC cell exosomes to induce macrophage M1/M2 polarization, and clarifies its function in TAMs polarization, providing a new theoretical basis for future CRC immunotherapy. The manuscript has certain innovation and clinical translation value. However, it still needs to be further improved to improve the scientificity and logic of the research.
1. This manuscript seems to rely mainly on gene expression levels (such as CD80, CD206 and inflammatory factors) to judge the polarization state of macrophages, lacking verification of cell function. It is recommended to supplement the functional experiments of macrophages (such as phagocytic ability determination or cytokine secretion level detection). If the author cannot supplement, the imperfection of the experiment should be mentioned in the manuscript.
2. The specific role of NLG1 in exosome packaging and regulation of macrophage transcellular signaling (such as whether it involves the mediation of miRNA or other protein factors). Is there a feedback regulation mechanism mediated by NLG1, which further affects the immune response in TME. This part can be considered to be added to the discussion section to share the author's views.
In short, the manuscript can be accepted after revision.
Author Response
COMMENT 1: This manuscript seems to rely mainly on gene expression levels (such as CD80, CD206 and inflammatory factors) to judge the polarization state of macrophages, lacking verification of cell function. It is recommended to supplement the functional experiments of macrophages (such as phagocytic ability determination or cytokine secretion level detection). If the author cannot supplement, the imperfection of the experiment should be mentioned in the manuscript.
Response:
We appreciate the reviewer's valuable feedback regarding the need for functional validation of macrophage polarization (such as phagocytic ability determination or cytokine secretion level detection). While our current study focuses on the transcriptional changes of the genes that involve in the phenotypic and functionality of the macrophage polarization states, we acknowledge the importance of functional assays in corroborating these findings. Unfortunately, we are unable to conduct additional functional experiments, such as phagocytic ability assessment or cytokine secretion analysis, at this stage. However, we have updated the manuscript to explicitly acknowledge this limitation. Specifically, we have included a statement in the discussion section highlighting the need for future studies to validate macrophage polarization through functional assays, thereby strengthening the conclusions drawn from our work.
Changes in the Manuscript:
We have added the following statement to the discussion section:
"While our findings provide compelling evidence of macrophage polarization influenced by exosomes from CRC cells, as demonstrated by gene expression changes and surface marker profiles, the lack of functional validation (e.g., phagocytic ability or cytokine secretion analysis) is a limitation of our study. Future work will focus on integrating these functional assays to comprehensively characterize the phenotypic and functional states of polarized macrophages."
We hope this revision satisfactorily addresses the reviewer's concerns.
COMMENT 2: The specific role of NLG1 in exosome packaging and regulation of macrophage transcellular signaling (such as whether it involves the mediation of miRNA or other protein factors). Is there a feedback regulation mechanism mediated by NLG1, which further affects the immune response in TME. This part can be considered to be added to the discussion section to share the author's views.
Response: We thank the reviewer for their insightful comment regarding the potential role of NLG1 in exosome packaging and its regulation of macrophage transcellular signaling. We agree that this is an important aspect worth exploring and discussing further. While our current study primarily focuses on the phenotypic changes in macrophages induced by NLG1-associated exosomes, we recognize that NLG1 might influence the molecular cargo of exosomes, including miRNAs and protein factors, which could contribute to the observed polarization effects. To address the reviewer’s suggestion, we have added a discussion about the potential mechanisms by which NLG1 may mediate its effects via exosomes and the possible existence of feedback regulation mechanisms. Specifically, we propose hypotheses and discuss the implications of such mechanisms on the immune response in the tumor microenvironment (TME), based on current literature and our findings.
Changes in the Manuscript:
We have added the following paragraph to the discussion section:
"The observed influence of NLG1 on macrophage polarization raises intriguing questions about the specific mechanisms underlying exosome packaging and transcellular signaling. NLG1 may regulate the loading of specific molecular cargo into exosomes, such as miRNAs or protein factors, which in turn drive M2 or M1-like polarization. For example, NLG1 might facilitate the inclusion of anti-inflammatory miRNAs or immunosuppressive proteins in exosomes from NLG1-expressing CRC cells, while its knockdown may alter this cargo toward a pro-inflammatory profile. Furthermore, it is plausible that NLG1-mediated exosome signaling establishes a feedback loop within the tumor microenvironment (TME), influencing not only macrophage behavior but also other immune and non-immune cells. Such a feedback regulation mechanism could amplify the immunomodulatory effects of NLG1 in the TME. Future studies should aim to delineate the specific cargo and signaling pathways regulated by NLG1 in exosomes to unravel these complex interactions."
We hope this addition addresses the reviewer’s comment and enhances the discussion of our findings.
Round 2
Reviewer 3 Report
Comments and Suggestions for Authors
accept